# Intestinal Goblet Cell Loss during Chorioamnionitis in Fetal Lambs: Mechanistic Insights and Postnatal Implications

**DOI:** 10.3390/ijms22041946

**Published:** 2021-02-16

**Authors:** Charlotte van Gorp, Ilse H. de Lange, Kimberly R. I. Massy, Lilian Kessels, Alan H. Jobe, Jack P. M. Cleutjens, Matthew W. Kemp, Masatoshi Saito, Haruo Usada, John Newnham, Matthias Hütten, Boris W. Kramer, Luc J. Zimmermann, Tim G. A. M. Wolfs

**Affiliations:** 1Department of Pediatrics, School of Oncology and Developmental Biology (GROW), Maastricht University, 6202 AZ Maastricht, The Netherlands; c.vangorp@maastrichtuniversity.nl (C.v.G.); i.delange@maastrichtuniversity.nl (I.H.d.L.); k.massy@maastrichtuniversity.nl (K.R.I.M.); lilian.kessels@maastrichtuniversity.nl (L.K.); matthias.hutten@mumc.nl (M.H.); b.kramer@mumc.nl (B.W.K.); luc.zimmermann@mumc.nl (L.J.Z.); 2Department of Surgery, School for Nutrition, Toxicology and Metabolism (NUTRIM), Maastricht University, 6202 AZ Maastricht, The Netherlands; 3Division of Neonatology/Pulmonary Biology, The Perinatal Institute, Cincinnati Children’s Hospital Medical Center, University of Cincinnati, Cincinnati, OH 4522, USA; alan.jobe@cchmc.org; 4Department of Pathology, School for Cardiovascular Diseases (CARIM), Maastricht University, 6202 AZ Maastricht, The Netherlands; jack.cleutjens@maastrichtuniversity.nl; 5Division of Obstetrics and Gynecology, The University of Western Australia, Crawley, WA 6009, Australia; matthew.kemp@uwa.edu.au (M.W.K.); masatoshi.saito.b4@tohoku.ac.jp (M.S.); haruwowwow777@yahoo.co.jp (H.U.); john.newnham@uwa.edu.au (J.N.); 6Center for Perinatal and Neonatal Medicine, Tohoku University Hospital, Sendai 980-8574, Miyagi, Japan; 7Department of Biomedical Engineering, School for Cardiovascular Diseases (CARIM), Maastricht University, 6202 AZ Maastricht, The Netherlands

**Keywords:** ER stress, perinatal, mucus barrier, intestinal inflammation

## Abstract

Chorioamnionitis, an important cause of preterm birth, is linked to necrotizing enterocolitis (NEC). NEC is characterized by a disrupted mucus barrier, goblet cell loss, and endoplasmic reticulum (ER) stress of the intestinal epithelium. These findings prompted us to investigate the mechanisms underlying goblet cell alterations over time in an ovine chorioamnionitis model. Fetal lambs were intra-amniotically (IA) exposed to lipopolysaccharides (LPS) for 5, 12, or 24 h, or 2, 4, 8, or 15 d before premature delivery at 125 d gestational age (GA). Gut inflammation, the number, distribution, and differentiation of goblet cells, ER stress, and apoptosis were measured. We found a biphasic reduction in goblet cell numbers 24 h–2 d after, and 15 d after IA LPS exposure. The second decrease of goblet cell numbers was preceded by intestinal inflammation, apoptosis, and crypt ER stress, and increased SAM-pointed domain-containing ETS transcription factor (SPDEF)-positive cell counts. Our combined findings indicated that ER stress drives apoptosis of maturating goblet cells during chorioamnionitis, ultimately reducing goblet cell numbers. As similar changes have been described in patients suffering from NEC, these findings are considered to be clinically important for understanding the predecessors of NEC, and targeting ER stress in this context is interesting for future therapeutics.

## 1. Introduction

Chorioamnionitis is an important risk factor for preterm birth and contributes to neonatal morbidity and mortality [1]. This antenatal condition, caused by intra-amniotic exposure to microorganisms and their toxic components (e.g., lipopolysaccharides (LPS)), is defined as inflammatory cell infiltration in the fetal membranes (chorion and amnion) and the amniotic cavity [1]. During chorioamnionitis, the fetus is exposed to an infectious environment and this can initiate a fetal inflammatory response syndrome (FIRS), which is characterized by elevated systemic interleukin (IL)-6 and IL-8 concentrations [2,3]. In addition, the contaminated amniotic fluid (AF) is swallowed by the fetus and thereby directly affects the fetal gut. From previous studies that used preclinical chorioamnionitis models, it is known that intrauterine inflammation/infection (caused by, e.g., LPS or *Ureaplasma* species) is associated with intestinal inflammation and concomitant developmental disruptions and gastrointestinal injury [4,5,6,7,8,9]. Accordingly, chorioamnionitis is a risk factor for adverse postnatal intestinal outcomes, including necrotizing enterocolitis (NEC) [10,11,12]. NEC is a severe intestinal disease that is characterized by intestinal inflammation and, in severe stages, necrosis [13]. Hence, NEC is associated with a high mortality rate of up to 50% [12,13].

Intestinal goblet cells produce and secrete mucins that are an important part of the mucus barrier, which forms the first passive line of defense against infiltration of bacteria and bacterial-derived products [14,15]. Importantly, in a murine study by Elgin et al., reduced goblet cell numbers were observed at birth as a result of maternal-induced fetal inflammation, indicating chorioamnionitis may already disrupt the mucus barrier [7]. In addition, neonates with NEC experience intestinal mucosal barrier disruption. This is characterized by a reduced number of mucin-2 (MUC2)-positive goblet cells and reduced production of mucins, which compromises the mucus barrier [16,17,18]. This disrupted mucus barrier allows bacteria to reach the intestinal epithelium and aggravate intestinal inflammation [18]. Intestinal goblet cells derive from intestinal stem cells that reside near the bottom of the intestinal crypt [19]. These stem cells constantly yield transit-amplifying daughter cells that proliferate and differentiate while migrating from the crypts into the lower part of the intestinal villi [20]. Once fully differentiated, intestinal goblet cells continue to migrate towards the villus tips where they finally die and shed into the intestinal lumen [20].

Intestinal epithelial endoplasmic reticulum (ER) stress is a mechanism involved in NEC pathophysiology that may underlie the loss of goblet cells in NEC [21,22]. During ER stress, unfolded proteins accumulate in the ER lumen and binding immunoglobulin protein (BiP) initiate an unfolded protein response (UPR) to resolve the unfolded protein synthesis [23,24]. If the UPR is unable to process the accumulation of unfolded proteins, apoptosis is induced through various pathways including the upregulation of transcription factor C/enhancer binding protein homologous protein (CHOP) [25]. Goblet cells are sensitive to ER stress due to their secretory nature and the large size of mucins that are folded in the ER [26,27,28]. ER stress in goblet cells typically arises during intestinal inflammation [25], which can trigger goblet cells to both synthesize and secrete mucins [27,29]. Post translation, mucins are transported to the ER for protein folding [25], and consequently, an increased mucin production during inflammation can lead to an accumulation of unfolded proteins in the ER and subsequent ER stress [28]. In human specimens from infants with NEC, UPR activation was associated with intestinal inflammation, increased morphological damage, and worse outcome [21]. In addition, knockout of important ER stress pathways prevented NEC development in a mouse model of NEC [22].

Currently, the time dependent changes and mechanisms that underlie goblet cell alterations following chorioamnionitis remain unknown. To address these mechanistic questions, we longitudinally studied the effects of chorioamnionitis by intra-amniotically (IA) administering LPS at different time points (5, 12, and 24 h, and 2, 4, 8, and 15 d) before premature delivery.

## 2. Results

### 2.1. Intestinal Inflammation

To investigate intestinal inflammation in fetal lambs after IA LPS exposure, myeloperoxidase (MPO)-positive cells were counted in the distal ileum (Figure 1). Compared to control fetuses (Figure 1A), the number of MPO-positive cells increased (*p* ≤ 0.05) at 4 and 8 d after IA LPS exposure (Figure 1B–D) and remained high until 15 d after LPS exposure (*p* = 0.06) (Figure 1D).

### 2.2. Goblet Cell Number and Distribution

To assess alterations in goblet cell numbers and distribution, Alcian blue (AB)/periodic acid–Schiff (PAS)- and mucin-2 (MUC2)-positive cells were identified in the distal ileum (Figure 2 and Figure 3). Compared to control animals (Figure 2A,D), the number of mature goblet cells, indicated by AB/PAS-positive cells, tended to be reduced at 24 h after (*p* = 0.10) (Figure 2B,D) and 2 d after (*p* = 0.07) (Figure 2D) IA LPS exposure. This reduction was normalized (*p* ≤ 0.05) in the 8-d LPS exposure group (Figure 2D). A second decrease (*p* ≤ 0.05) in the number of goblet cells was observed at 15 d after IA LPS exposure compared to control (*p* = 0.09) and the 8-d LPS exposure group (*p* ≤ 0.05) (Figure 2C,D).

No statistically significant group differences were found with regard to total MUC2-positive cell counts (Appendix A). Interestingly, the presence of AB/PAS-positive cells largely paralleled that of MUC2-positive cells in the villi tips; the number of MUC2-expressing cells tended to be increased in the villi tips of the distal ileum (*p* = 0.09) at 8 d after IA LPS exposure compared to 24 h after IA LPS exposure (Figure 3A,B,D). At 15 d after IA LPS exposure, the number of MUC2-positive cells tended to be reduced (*p* = 0.10) compared to 8 d after IA LPS exposure (Figure 3C,D). No differences were observed in MUC2-positive cell counts in ileal crypts, lower villi, and middle villi.

### 2.3. Intestinal Epithelial Cell Death

As apoptosis is associated with intestinal inflammation [30] and may explain the loss of goblet cells following chorioamnionitis, intestinal epithelial apoptosis (cleaved-caspase 3 (CC3)) was studied (Figure 4). The number of CC3-positive cells was elevated from 2 d after IA LPS exposure onwards and normalized at 15 d after IA LPS exposure, with the highest count at 4 d after IA LPS exposure compared to controls (*p* < 0.05) (Figure 4A–D). Apoptosis was mainly detected in the lower villi region (Figure 4B). Increased cell death did not cause villus atrophy (Appendix A), which indicates loss of goblet cells relative to enterocytes following IA infection.

### 2.4. Goblet Cell Differentiation

In addition to apoptosis, goblet cell differentiation was studied as a potential mechanism that underlies the reduced number of goblet cells. SAM-pointed domain-containing ETS transcription factor (SPDEF)-positive cells, counterstained with AB/PAS were counted in all groups (Figure 5). The number of SDPEF-positive cells was increased at 4 d after IA LPS exposure (*p* = 0.09) and this increase reached statistical significance at 8 d after IA LPS exposure compared to controls (*p* ≤ 0.05) (Figure 5B,D). At 15 d after IA LPS exposure, the SPDEF-positive cell count still tended to be increased (*p* = 0.06) (Figure 5C,D). For all time points, most SPDEF-positive cells were observed in the lower villi region.

### 2.5. ER-Stress in Distal Ileum of Premature Lambs

To study the underlying mechanism behind alterations in goblet cell numbers, differentiation, and distribution, the BiP expression pattern and CHOP-positive cell counts in the ileal crypt-villi axis were studied (Figure 6). BiP is an ER chaperone protein that under physiological conditions prevents inositol-requiring protein 1α (IRE1α) signaling [24]. However, under ER stress, a condition which secretory cells including goblet cells are especially vulnerable to, BiP accumulates and dissociates from IRE1α, thereby initiating the unfolded protein response [24]. In control fetuses, BiP immunoreactivity is mainly present in the lower villi region of the crypt-villus axis (Figure 6A,D). The BiP expression pattern changed at 4 d after IA LPS exposure, where BiP immunoreactivity was detected in the ileal crypts (Figure 6B,D). Interestingly, BiP expression pattern was also changed in the 15 d after LPS group. In contrast to the crypt expression at 4 d after IA LPS exposure, the 15-d pattern was characterized by increased BiP immunoreactivity in the villi tips (Figure 6C,D).

CHOP is associated with ER stress-induced apoptosis, which indicates progression of ER stress [25]. The number of CHOP-positive cells was increased in the whole crypt-villus axis following IA LPS, but predominantly highest in the intestinal crypt (Figure 7A–E). Increased CHOP-positive cell counts in the intestinal crypt were present from 5 h after IA LPS exposure onwards, with the highest CHOP-positive cell numbers at 12 h (*p* < 0.05) (Figure 7B,D) and 2 d (*p* = 0.08) (Figure 7F) after IA LPS exposure compared to controls. At 4 d after IA LPS exposure, CHOP-positive cell counts in the intestinal crypt were normalized (Figure 7C,F). Interestingly, a second increase of CHOP-positive cell numbers was observed at 8 (*p* = 0.08) (Figure 7F) and 15 d (*p* < 0.05) after IA LPS exposure (Figure 7D,F).

## 3. Discussion

In the current study, we investigated the effects of LPS-induced chorioamnionitis on the number, distribution, and differentiation of goblet cells. In addition, we studied whether intestinal inflammation, intestinal epithelial ER stress and apoptosis are mechanistically involved.

LPS-induced chorioamnionitis was observed to cause a biphasic decrease in goblet cell counts. The first reduction coincided with early systemic inflammation 24 h after IA LPS exposure (IL6) [2] and intestinal inflammation at 24 h (IL8) [31] and 2 d (macrophages) (Heymans et al. submitted for publication) after IA LPS exposure that we observed in earlier published studies of the same experiment. It is therefore probably caused by emptying of goblet cell secretory vesicles as a natural response to acute inflammation [29]. Secretion of mucins by the goblet cells will result in decreased volume of the goblet cell theca and therefore can reduce goblet cell counts on analysis of the AB/PAS staining. While it could alternatively be due to goblet cell loss, the unaltered SPDEF immunoreactivity and the lack of apoptosis within this period do not support this scenario. Of note, the number of SPDEF-positive cells was higher than the number of AB/PAS-positive cells in all experimental groups. Several factors could explain this apparent discrepancy. First, mucus secretion lowers the amount of AB/PAS-positive cells counted. Second, developing goblet cells, which already express SPDEF [32], have no theca or small theca sizes and, thus, are likely to be AB/PAS-negative. This latter notion is supported by the high SPDEF-positive cell counts in the lower villi region and could be related to normal intestinal development of the fetus. Future studies are needed to discern why a relatively large amount of cells in the lower villi region express SPDEF in utero.

Interestingly, the initial reduction in goblet cell numbers was restored at 8 d after IA LPS exposure, indicating that the effects of acute inflammation on goblet cell numbers were compensated. This compensation could be explained by the increase in goblet cell differentiation (increased SPDEF immunoreactivity) at 4 and 8 d after IA LPS exposure that is most likely triggered by inflammation. Accordingly, intraperitoneal administration of the pro-inflammatory cytokine TNFα caused an increase in mRNA level of the goblet cell differentiation marker SPDEF in neonatal mice [17]. In addition, the pro-inflammatory cytokine IL-1β induced SPDEF mRNA levels in human bronchial epithelial cells [33]. Moreover, the IL-1β-induced increase in MUC5 protein expression and secretion, the main mucin of lung goblet cells, was strongly reduced in SPDEF-deficient mice [33], which suggests that SPDEF has an important role in upregulating mucus production and secretion in mucosal tissue upon inflammation.

Importantly, at 15 d after IA LPS exposure, the reduction of AB/PAS-positive cell counts in combination with reduced numbers of MUC2 positive cells at the villus tips indicates that a loss of mature goblet cells was involved in the second wave of reduced goblet cell counts. Moreover, the absence of villus atrophy strongly suggests this loss was goblet cell-specific (e.g., greater loss of goblet cells than enterocytes). Three scenarios might be responsible for the observed loss of mature goblet cells: First, chorioamnionitis-driven intestinal inflammation might induce cell death of mature goblet cells. However, since in the villus tips neither apoptosis nor epithelial damage was present, this scenario is not likely. Second, disturbed goblet cell differentiation by altered programming of maturating goblet cells may be involved. Yet, this concept is not supported by the finding that the number of SPDEF-positive cells was not decreased 15 d after IA LPS exposure. Finally, cell death of maturating cells that prevents the formation of mature goblet cells could be responsible. This latter concept is supported by our findings of increased apoptosis in the lower villus region, since apoptosis is often present in the crypt and in the villi tips, but it rarely occurs in the lower villi [34]. Collectively, our combined findings indicate that prolonged antenatal LPS exposure triggers a loss of mature goblet cells through cell death of maturating goblet cells, potentially predisposing to postnatal pathologies.

The increase in intestinal inflammation and intestinal epithelial ER stress in the crypts provide insight into the potential mechanisms behind the apoptosis in the lower villus region. Mild ER stress is important for the development of secretory cells [35]. Consequently, BiP expression is naturally present in the lower villus region [36], which is important for terminal differentiation of goblet cells [19]. However, from 5 h until 4 d after IA LPS exposure, ER stress emerged in the intestinal crypts. The identified ER stress in the crypts was considered to be injurious, as a study by Afrazi et al. reports ER stress in LGR5-positive stem cells to be associated with apoptosis within the intestinal crypt and an increased NEC severity [22]. Besides stem cells, Paneth cells, which are vital for stem cell maintenance, may be subject to crypt ER stress [37]. Thus, ER stress in crypt intestinal epithelial cells can either directly or through stem cell dysfunction affect maturating goblet cells and result in apoptosis in the lower villus region following migration from the crypts. We consider inflammation to be the driving force behind ER stress and subsequent intestinal epithelial apoptosis, leading to reduced numbers of mature goblet cells. Intestinal inflammation and ER stress are bidirectionally linked. Intestinal inflammation can induce ER stress [25], and intestinal epithelial ER stress can aggravate intestinal inflammation, indirectly through disturbance of the mucus layer and Paneth cell secretion [25] and directly by UPR-initiated inflammatory signals [25,38]. Thus, in the current study, we proposed—as an underlying mechanism of the observed loss of mature goblet cells—a vicious circle of inflammation and ER stress that leads to apoptosis of maturating goblet cells.

Importantly, similar changes have been described in patients suffering from NEC [16,17,18,22], which suggests that a loss of goblet cells in utero may predispose to NEC development. Reduced goblet cell numbers are associated with a disturbed mucus layer [39] and, thus, a reduced intestinal barrier function, which makes a neonate upon birth more vulnerable during the crucial period of microbial colonization. In addition, the increase of CHOP-positive cells in the crypt at 8–15 d after IA LPS exposure implies that the intestine is prone to a second wave of apoptosis, either following ongoing (intrauterine) inflammation or after additional postnatal inflammatory hits such as mechanical ventilation [40] or sepsis [41]. This concept is supported by recent findings in a murine chorioamnionitis model in which chorioamnionitis not only induced goblet cell loss in utero, but also predisposed to further reduced goblet cell numbers following a postnatal inflammatory hit [7].

Of note, several nutritional interventions, such as epidermal growth factor (EGF), heparin-binding EGF-like growth factor, insulin-like growth factor 1, human milk oligosaccharides (HMO), and bovine and human breast milk exosomes, have been shown to influence goblet cells and mucus barrier function in animal NEC models [42,43,44,45,46,47]. ER stress may be a crucial mechanism involved. Enteral administration of bovine milk exosomes reduced intestinal injury and inflammation, and prevented a loss of goblet cells in a murine NEC model [45]. Importantly, treatment with these bovine milk exosomes also increased the number of cells positive for GRP94 [45], an ER chaperone protein that is important for goblet cell maintenance [48]. Likewise, supplementation of formula with human milk oligosaccharides (HMO) reduced murine NEC injury and prevented a loss of goblet cells, concomitant with an increased protein expression of the ER chaperone proteins BiP and protein disulfide isomerase (PDI) [47]. Moreover, administration of the PDI inhibitor rutin abolished the HMO effects on intestinal injury and goblet cell numbers, which shows that ER stress plays a crucial role in the pharmacological effect of HMO in NEC [47]. We have previously shown that a nutritional intervention with plant sterols can already positively affect the gut outcome in chorioamnionitis in utero [8]. Targeting in utero alterations of the mucus barrier with nutritional interventions is therefore a promising topic for future research.

An important strength of the current study is the longitudinal study set-up, which allowed us to gain mechanistic insight regarding the goblet cell alterations observed. A limitation is the low number of animals per treatment group, which is inherent to the large animal model used. In addition, our study set-up did not rule out an effect of gestational age at the start of IA LPS exposure.

In summary, we reported a biphasic reduction in goblet cell counts after IA LPS exposure. Whereas the first reduction wave could be explained by mucin secretion, the second resulted from loss of mature goblet cells that was strongly linked with intestinal inflammation, ER stress, and concomitant apoptosis. Our combined findings indicate that ER stress-driven apoptosis of maturating goblet cells gives rise to reduced goblet cell numbers in chorioamnionitis. As in utero goblet cell loss has been experimentally shown to persist postnatally and predisposes to further disruption of goblet cells following additional inflammatory hits [7], chorioamnionitis-driven goblet cell loss may play a role in NEC pathophysiology and potentially in other adverse gut outcomes. In this context, ER stress is an interesting target for future therapeutics.

## 4. Material and Methods

### 4.1. Experimental Design

Animal experiments were approved by the Animal Ethics/Care Committee of the University of Western Australia (Perth, Australia; ethical approval number: RA/3/100/928; approval date 1 January 2010). The study design was previously published [2,49]. Briefly, singleton fetuses from time-mated merino crossbred ewes were randomly assigned to eight study groups. IA saline (controls; 0.9% saline solution) or IA LPS injections (10 mg *Escherichia coli*-derived LPS; O55:B5; Sigma–Aldrich, St. Louis, MO, USA) were performed under ultrasound guidance at 5, 12, or 24 h, or 2, 4, 8, or 15 d before preterm delivery at 125 d gestational age (GA), which is for the gut comparable to approximately 27–30 weeks of human GA (term ~150 d GA) (Figure 8). IA LPS administration has previously been shown to induce inflammation of the chorion and amnion and influx of inflammatory cells in the amniotic fluid that correlates well with human chorioamnionitis [50]. The half-life time of IA-administered LPS is relatively long (1.7 days) and the LPS remains detectable until 15 d after IA injection [50]. Because no differences between various durations of saline exposure were found within the control group, all the saline animals were pooled into one control group. Directly after preterm delivery (125 d GA), the lambs were euthanized by intravenous injection of pentobarbitone (100 mg/kg, Valaber, Pitman–Moore, Australia). Distal ileum tissue was collected post-mortem for analyses. The involved investigators were blinded to treatment allocation during data analyses.

### 4.2. Antibodies

The following antibodies were used: Polyclonal rabbit antibody against human myeloperoxidase (MPO) (A0398, Dakocytomation, Glostrup, Denmark); polyclonal rabbit anti-muc2c3 (kindly provided by the University of Gothenburg, Gothenburg, Sweden); polyclonal rabbit anti-SAM-pointed domain containing ETS transcription factor (SPDEF) (ab197375, abcam, Cambridge, United Kingdom); monoclonal rabbit anti-BiP (C50B12, cell signaling technology, Leiden, The Netherlands); monoclonal mouse anti-CHOP (clone 9C8, MA1-250, Thermo Fisher Scientific, Waltham, MA, USA), and polyclonal rabbit anti-cleaved caspase-3 (CC3) (asp175) (9661, cell signaling technology, Leiden, The Netherlands).

Secondary antibodies used were: Peroxidase-conjugated goat anti-rabbit (111-035-045, Jackson, West Grove, PA, USA), biotin-conjugated rabbit anti-mouse (E0413, DakoCytomation, Glostrup, Denmark), and biotin-conjugated swine anti-rabbit (E0353, DakoCytomation, Glostrup, Denmark).

### 4.3. Alcian Blue/Periodic Acid–Schiff (AB/PAS)

AB/PAS was used as a general marker for mature (e.g., glycosylated) mucins [51,52]. Distal ileum tissue was fixated in 4% paraformaldehyde and embedded in paraffin and 4-µm sections were cut with a microtome (Leica RM2235). Sections were stained with Alcian blue (8GC work-solution, 66011-500ML-F, Sigma–Aldrich, St. Louis, MO) for 30 min at room temperature (RT) followed by incubation in periodic acid (PA, 0.5%) for 5 min. After washing, slides were stained with Schiff’s reagent (3952016-500ML, Sigma–Aldrich, St. Louis, MO) for 15 min. Stained slides were scanned using Ventana IScan (Ventana Medical System, Inc., Tucson, AZ, USA). The number of AB/PAS-positive cells was manually counted with the use of Ventana Image Viewer (v3.1.4, Ventana Medical System, Inc., Tucson, AZ), and the number of positive cells per mm^2^ of ileal surface area was calculated after ileal surface area measurements with the use of Qwin Pro software (v3.4.0, Leica Microsystem, Wetzlar, Germany).

### 4.4. Immunohistochemistry

MPO was used as a marker for inflammation. MUC2 immunoreactivity was analyzed as an additional goblet cell marker; the antibody used detects both immature (not O-glycosylated) and mature (O-glycosylated) mucins. SPDEF was used to assess goblet cell differentiation [32,53,54]. BiP and CHOP staining were performed to study intestinal epithelial ER stress and CC3 to assess apoptosis. After tissue fixation in 4% paraformaldehyde and paraffin embedding, 4-µm thick ileal sections were cut and were used to perform immunohistochemical studies. Endogenous peroxidase activity was blocked with 0.3% H_2_O_2_ dissolved in phosphate-buffered saline (PBS) for 20 min. For MUC2, SPDEF, BiP, CHOP, and CC3, antigen retrieval was applied by boiling the slides in 10 mM sodium citrate buffer (pH 6.0). Aspecific binding sites were blocked with 4% normal goat serum (MPO, SPDEF) or 5% bovine serum albumin (MUC2, BiP, CHOP, CC3) for 30 min at room temperature (RT). Ileal sections were incubated with primary antibody for 1 h at RT (MPO) or incubated overnight at 4 °C (MUC2, SPDEF, BiP, CHOP, and CC3). Subsequently, slides were incubated with peroxidase-conjugated (MPO) or biotin-conjugated (MUC2, SPDEF, BiP, CHOP, and CC3) secondary antibodies at RT. MPO-positive cells were visualized using 3-amino-9-ethylcarbazole (AEC, Sigma–Aldrich, St. Louis, MO, USA); MUC2 and CC3 were detected with nickel-3, 3′–diaminobenzidine (NiDAB, Sigma Aldrich, St. Louis, MO, USA), and SDPEF, BiP, and CHOP were visualized using 3, 3′–diaminobenzidine (DAB, Sigma Aldrich, St. Louis, MO, USA). (Nuclear) counterstaining was performed with hematoxylin (MPO), nuclear fast red (MUC2 and CC3), or AB/PAS (SPDEF).

For MPO, SPDEF, MUC2, CHOP, and CC3, positive cells were manually counted in two ileal cross sections with the use of Ventana Image Viewer (v3.1.4, Ventana Medical System, Inc., Tucson, AZ) and corrected for ileal surface area with the use of Qwin Pro software (v3.4.0, Leica Microsystem, Wetzlar, Germany). Only good cross-sections through the full crypt-villus axis were used. The number of cells was depicted as number of cells per mm^2^ ileal surface area and the average count per sheep was presented. Because (developing) goblet cells migrate from the crypt bottom towards the villus tips over time [20], we considered the anatomical location of goblet cells and ER stress of importance for understanding the underlying mechanisms involved. We therefore analyzed the MUC2-positive cell count and CHOP-positive cell count in ileal crypts, the lower villi, the middle villi, and the tips of the villi, and corrected this for the adequate tissue surface area (Appendix A). Ileal crypts were selected based on tissue morphology. The ileal villi were divided in three parts (lower villi, middle villi, and villi tips) that were all one third of the average villus length. For BiP, staining intensity was also analyzed in the ileal crypts, the lower villi, the middle villi, and the tips of the villi (Appendix A) in two ileal cross sections with the following scoring system: No staining was scored as 0, mild staining was indicated as 1, moderate staining was scored as 2, and intense staining intensity was indicated with a score of 3.

### 4.5. Statistical Analysis

Statistical analyses were performed with GraphPad Prism Software (v6.0, Graphpad Software INc., La Jolla, CA, USA). Data are displayed as median with interquartile range (IQR) for all readouts. A Kruskal–Wallis test followed by Dunn’s post hoc test was used to test statistically significant differences between the different groups. Differences were interpreted as statistically significant at *p* ≤ 0.05. Given the relatively small number of animals per group, *p*-values between 0.05 and 0.10 were reported in exact numbers and were regarded as potentially biologically relevant, as previously described [2,8,55]. This assumption decreased the chance of a false negative finding, but increased the chance of false positive results.

## Figures and Tables

**Figure 1 ijms-22-01946-f001:**
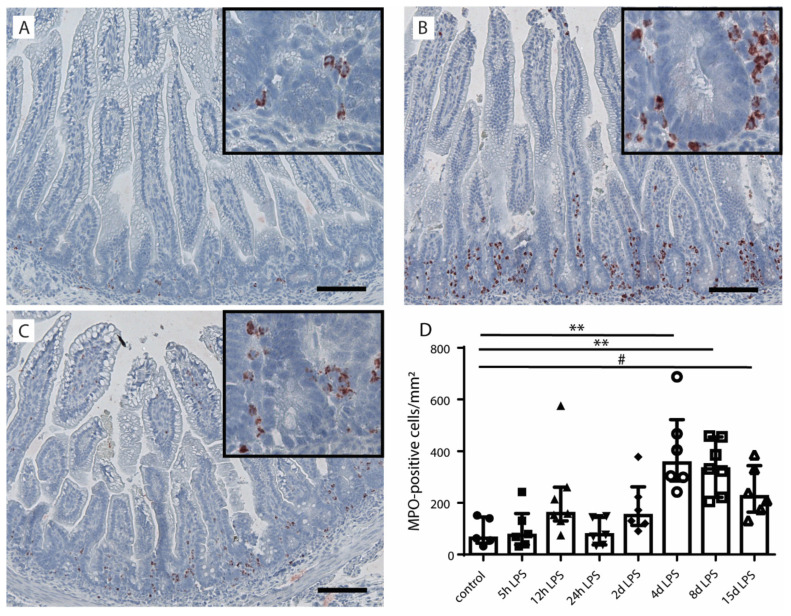
The number of myeloperoxidase (MPO)-positive cells per mm^2^ ileal surface area in the distal ileum of fetal lambs after intra-amniotic (IA) lipopolysaccharides (LPS) exposure. Few MPO-positive cells were detected in the distal ileum of control fetuses (**A**,**D**). The number of MPO-positive cells increased at 4 (**B**,**D**) and 8 d (**C**,**D**) and still tended to be increased at 15 d after IA LPS exposure (**D**). In all groups, MPO-positive cells were mainly detected near the intestinal crypts. Each data point represents the MPO-positive cell count of one lamb. Data are displayed as median with interquartile range. Scale bars indicate 100 µm. ** *p* < 0.01; ^#^ 0.05 < *p* ≤ 0.10. Abbreviations: MPO, myeloperoxidase.

**Figure 2 ijms-22-01946-f002:**
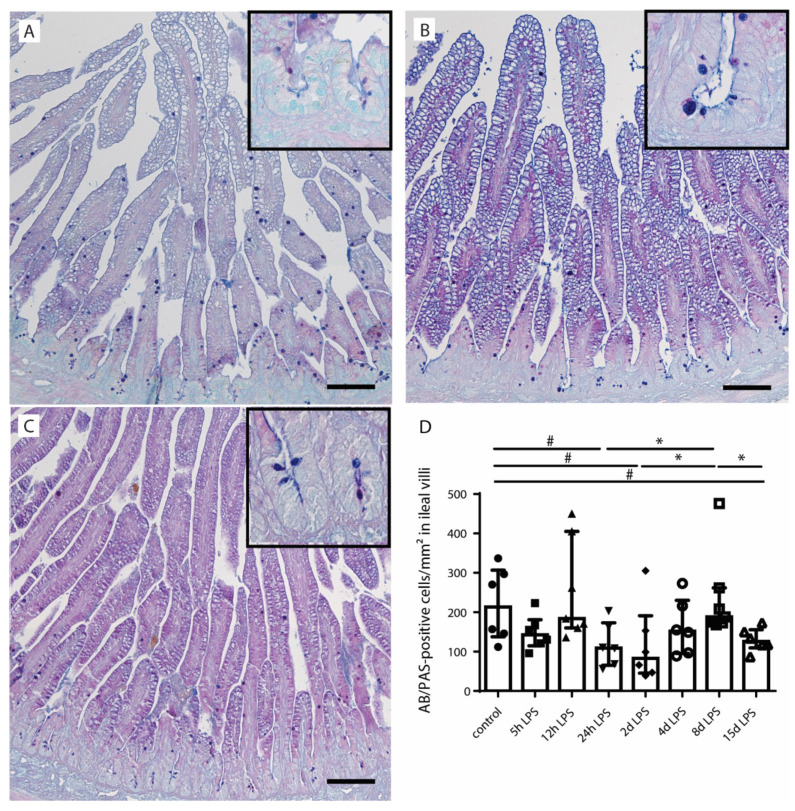
The number of Alcian blue (AB)/periodic acid–Schiff (PAS)-positive cells per mm^2^ surface area in distal ileum of premature lambs. The number of AB/PAS-positive cells in control lambs (**A**). This number tended to be decreased at 24 h (**B**,**D**) and 2 d (**D**) after IA LPS exposure. The number of AB/PAS-positive cells was normalized at 8 d after IA LPS exposure (**D**). However, a second reduction in the number of AB/PAS-positive cells was observed 15 d after LPS exposure (**C**,**D**). Each data point represents the AB/PAS-positive cell count of one lamb. Data are displayed as median with interquartile range. Scale bars indicate 100 µm. * *p* ≤ 0.05, ^#^ 0.05 < *p* ≤ 0.10. Abbreviations: AB/PAS; Alcian blue/periodic acid–Schiff.

**Figure 3 ijms-22-01946-f003:**
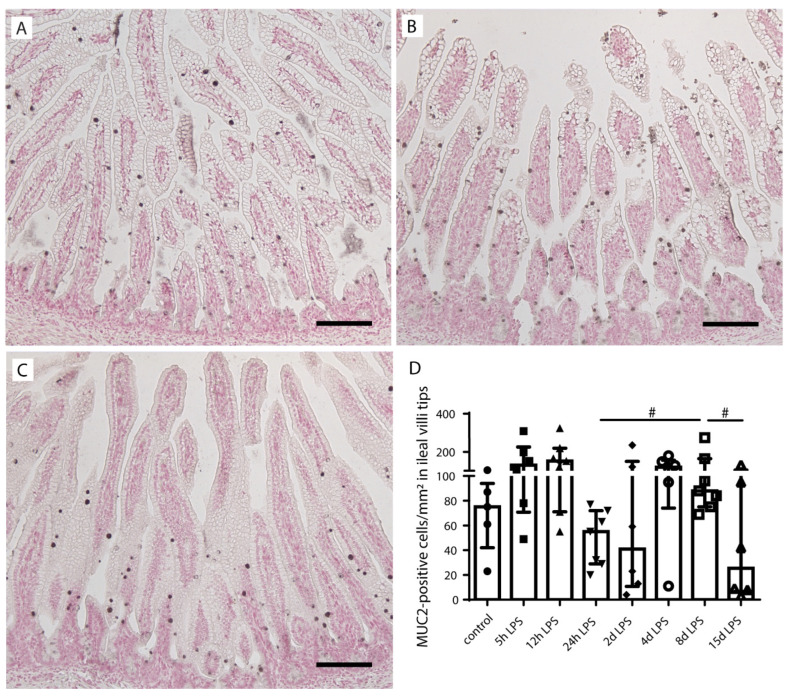
Mucin-2 (MUC2)-positive cell count in ileal villi tips of premature lambs. MUC2-positive cells in the distal ileum of control lambs (**A**,**D**). MUC2-positive cells tended to be increased in the 8 d after LPS group (**B**) compared to the 24 h after LPS exposure group (**D**). This increase in the 8-d LPS group tended to be reduced at 15 d after LPS exposure (**C**,**D**). MUC2-positive goblet cells were counted in the villi tips, and the count was corrected for ileal villi tip surface (cells/mm^2^) for all treatment groups. Each data point represents the MUC2-positive cell count of one lamb. Data are displayed as median with interquartile range. Scale bars indicate 100 µm. ^#^ 0.05 < *p* ≤ 0.10. Abbreviations: MUC2; mucin-2.

**Figure 4 ijms-22-01946-f004:**
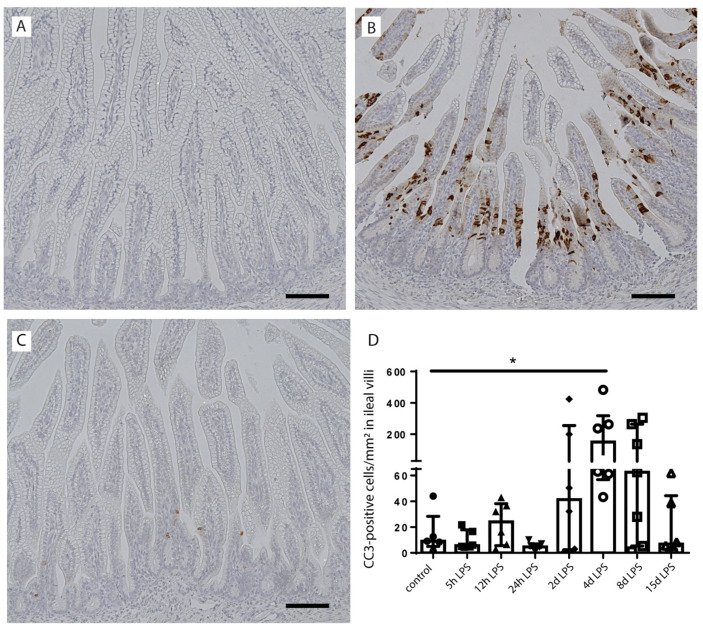
Presence of intestinal epithelial cell death in the distal ileum of premature lambs. In the control group, low counts of cleaved caspase 3 (CC3)-positive cells were found (**A**,**D**). The number of CC3-positive cells was statistically significantly increased at 4 d after IA LPS exposure compared to controls (**B**,**D**). At the other timepoints, including 15 d after IA LPS (**C**), no statistically significant increase in apoptosis was observed (**D**). The number of CC3-positive cells was corrected for ileal surface (cells/mm^2^) for all treatment groups. Each data point represents the CC3-positive cell count of one lamb. Data are displayed as median with interquartile range. Scale bars indicate 100 µm. * *p* ≤ 0.05. Abbreviations: CC3; cleaved-caspase 3.

**Figure 5 ijms-22-01946-f005:**
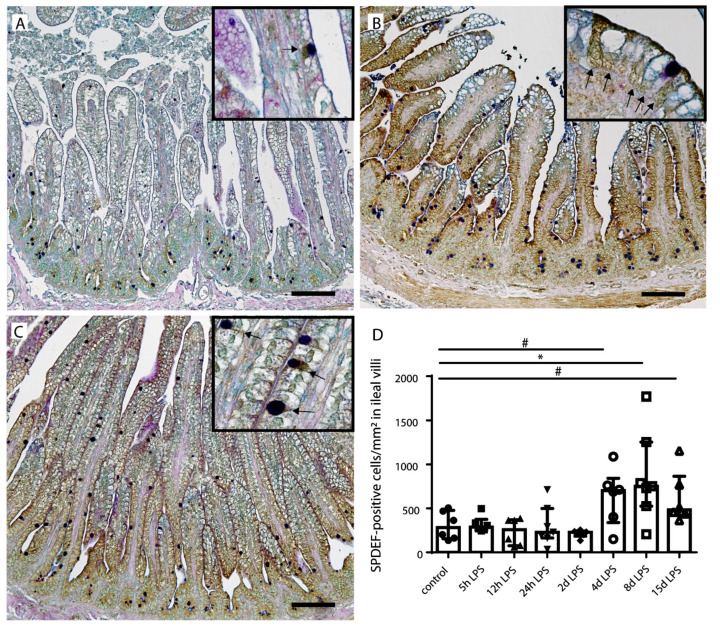
Number of SAM-pointed domain-containing ETS transcription factor (SPDEF)-positive goblet cells per mm^2^ surface area. In control fetuses, some SPDEF positive goblet cells were detected (**A**,**D**). The number of SDPEF-positive goblet cells (indicated by black arrows in inserts) was statistically significantly increased at 8 d after (**B**,**D**), and still tended to be increased at 15 d after (**C**,**D**) IA LPS exposure compared to the controls. Each data point represents the SPDEF-positive cell count of one lamb. Data are displayed as median with interquartile range. Scale bars indicate 100 µm. * *p* ≤ 0.05; ^#^ 0.05 < *p* ≤ 0.10. Abbreviations: SPDEF; SAM-pointed domain.

**Figure 6 ijms-22-01946-f006:**
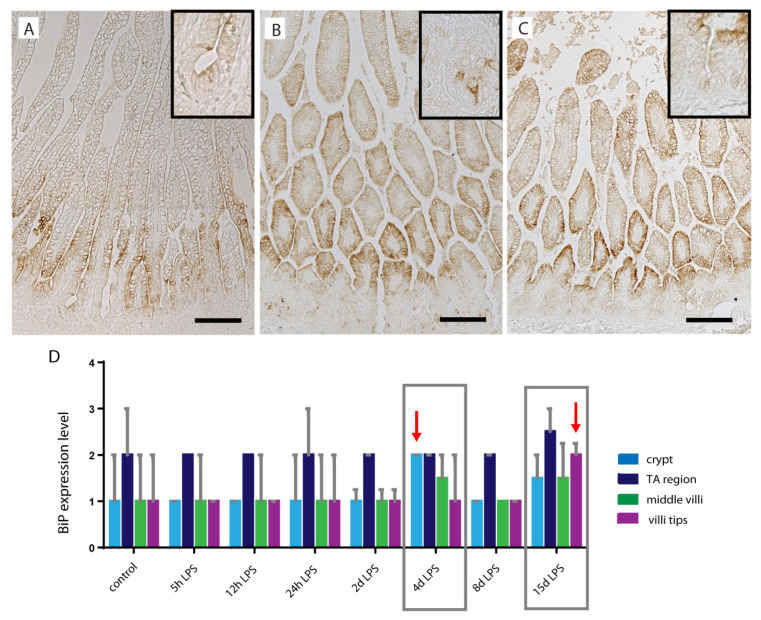
Binding immunoglobulin protein (BiP) expression pattern along crypt-villus axis in distal ileum of premature lambs. BiP was mainly expressed in the lower villi region, localized above the crypts, in the control fetuses (**A**,**D**). This expression pattern was changed in the 4 d after LPS group when BiP immunoreactivity was increased in the ileal crypts (indicated by arrow) (**B**,**D**), and at 15 d after IA LPS exposure, when BiP immunoreactivity was increased in the ileal villi tips (indicated by arrow) (**C**,**D**). BiP expression pattern was scored as follows: 0 = no BiP expression; 1 = mild BiP expression; 2 = moderate BiP expression; and 3 = intense BiP expression for all treatment groups. Data are displayed as median with interquartile range. Scale bars indicate 100 µm. Abbreviations: BiP; binding immunoglobulin protein.

**Figure 7 ijms-22-01946-f007:**
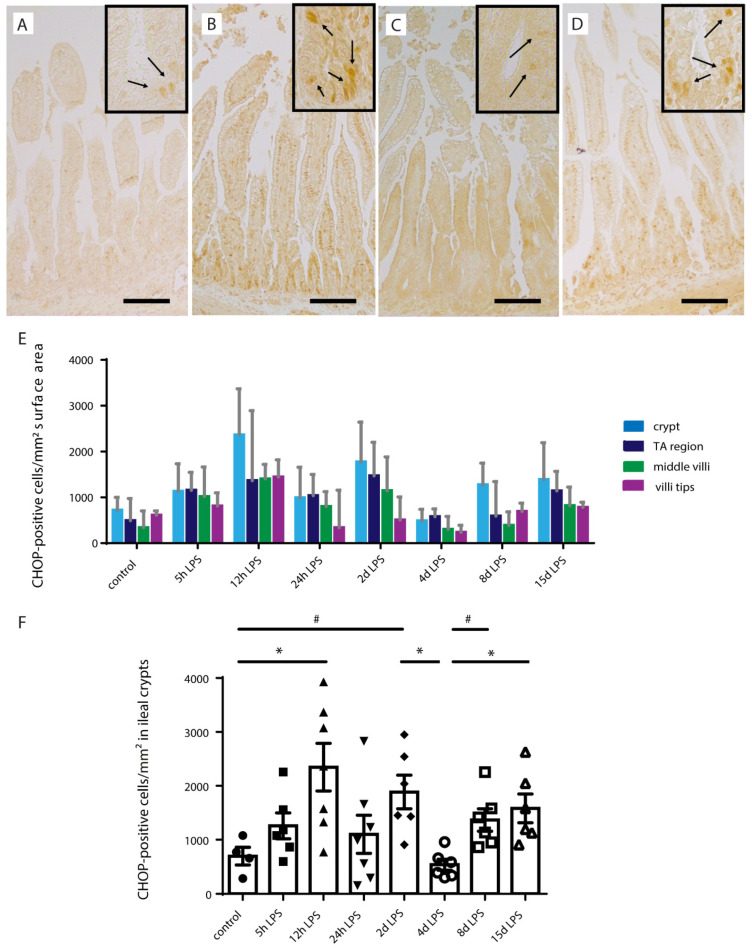
Number of transcription factor C/enhancer binding protein homologous protein (CHOP)-positive intestinal epithelial cells per mm^2^ surface area. In all groups, the number of CHOP-positive intestinal epithelial cells (indicated by black arrows in inserts) per mm^2^ surface area was highest in the intestinal crypts (**A**–**E**). From 5 h onwards, the number of CHOP-positive cells was elevated, with the highest counts at 12 h (*p* < 0.05) (**B**,**F**) and 2 d (*p* = 0.08) (**F**) after IA LPS exposure compared to controls. At 4 d after IA LPS exposure, the CHOP-positive intestinal epithelial cell counts normalized (**C**,**F**). A second increase in CHOP-positive intestinal epithelial cell count was observed at 8 (*p* = 0.08) (**F**) and 15 d (*p* < 0.05) after IA LPS exposure (**D**,**F**). Each data point represents the CHOP-positive cell count of one lamb. Data are displayed as median with interquartile range. Scale bars indicate 100 µm. * *p* ≤ 0.05; ^#^ 0.05 < *p* ≤ 0.10. Abbreviations: CHOP; transcription factor C/enhancer binding protein homologous protein.

**Figure 8 ijms-22-01946-f008:**
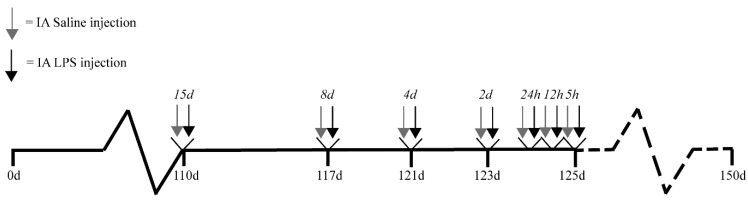
Study design. Singleton fetuses from time-mated merino crossbred ewes were randomly assigned to eight different study groups. IA saline or LPS (10 mg *Escherichia coli*-derived LPS) injections were given at seven different time points (5, 12, or 24 h, or 2, 4, 8, or 15 d) before preterm delivery at 125 d GA (~150 d GA is term). Saline exposed animals were pooled within one control group. Abbreviations: IA, intra-amniotic; GA, gestational age.

## Data Availability

The data presented in this study are available in this article and in the Appendix A.

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
