# Peer review of "Intestinal Goblet Cell Loss during Chorioamnionitis in Fetal Lambs: Mechanistic Insights and Postnatal Implications"

_ijms, 2021, doi:10.3390/ijms22041946_

Round 1

Reviewer 1 Report

The manuscript by van Gorp et al reports the findings of a single experiment in a large animal (sheep) model of chorioamnionitis and attempts to carefully monitor induced changes in the small intestinal goblet cells.  Goblet cell loss has been documented in equivalent models in rodents but this is the first report in a more clinically relevant model.  Seven different exposure times of LPS introduced into the amniotic fluid to allow monitoring of longitudinal changes in goblet cell populations prior to a pre-term delivery.  The strength of the study is the large animal model while weaknesses include sample size relative to the large variability in measures at some time points, histological analytic techniques and the quality of some of the protein stains, and the lack of analysis of post-birth infectious or inflammatory challenges that would enhance relevance to NEC.  I largely agree with the conclusions that have been made, and a sensible discussion is provided, however I have some doubt on the robustness of some of the data.

Some specific areas to consider to strengthen the study include:

  1. The number of sheep per group is not clearly stated in Materials & Methods or the Figure legends. I presume each data point is from one sheep with data pooled from the 2 sections analysed – this needs clarification.
  2. For the ileal cross-sections how were the 2 section areas selected – were only good cross-sections through crypt-villous axes used? Is 2 sections sufficient – ie. how close was the concordance between these sections?  The quality of the histological images and sections selected to demonstrate do not fill me with confidence in the robustness, perhaps contributing to the wide variation in findings at some time points.
  3. The data on goblet cell numbers based on Alcian blue staining (eg. Fig 3 data) will be influenced potentially by goblet cell secretion as larger theca (low secretion) are more likely to appear in sections. This is partially acknowledged in the Discussion of findings but needs to be explicitly acknowledged.  It would have been useful to have goblet cells separately counted in the crypts and villi.
  4. The SPDEF staining is not convincingly specific, all goblet cells should be SPDEF positive so one would predict it should really replicate the Alcian blude data – however, if relying on seeing the theca and SPEDF staining the sectioning angle through individual cells will be impactful. Was there an increase of goblet cell precursors in the base of the crypts?
  5. While I agree there is a high density of CHOP positive cells in the crypt, there are large numbers of villous cells (too numerous for gc) with clear nuclear staining in Fig 8B and D (CHOP is a transcription factor) and this should be described.
  6. The CC3 apoptosis staining is described as being in the transit amplifying cells but looks to be too high in the crypt-villous axis to be in these cells. An alternative possibility is that these are newly differentiating goblet cells.  This is an important distinction that requires further exploration.

Suggested corrections

  1. Line 268 - BiP is a ubiquitous ER chaperone so is present in health but accumulates in ER stress and it is its disassociation from IRE1 that initiates the UPR.Focus should be on gc.
  2. Lines 309-313 – not clear here what is description of current findings and what is being cited from literature – its inappropriately muddled
  3. Line 325-27 – SPDEF has critical role in mucus production in health and inflammation driving over 100 genes involved in mucin biosynthesis and secretion
  4. Fig 5 shows apoptosis – claimed to be in TA cells but looks too high in the villi
  5. line 194 “of” should be “after” as single LPS exposure with unknown half life?
  6. Line 120 – spelling “Cambridge”
  7. Line 156 – spelling “biotin”

Author Response

Dear reviewers,

Thank you very much for offering us the opportunity to submit a revised version of our manuscript IJMS-1090160 entitled: "Intestinal goblet cell loss during chorioamnionitis in fetal lambs: mechanistic insights and postnatal implications”

We thank the reviewers for their interest and appreciation of our work. We are grateful for the valuable comments that we feel have improved our manuscript.

Please find enclosed a revised version of the manuscript and a point to point response to the comments of the reviewers.

Reviewer 1:

The manuscript by van Gorp et al reports the findings of a single experiment in a large animal (sheep) model of chorioamnionitis and attempts to carefully monitor induced changes in the small intestinal goblet cells. Goblet cell loss has been documented in equivalent models in rodents but this is the first report in a more clinically relevant model.  Seven different exposure times of LPS introduced into the amniotic fluid to allow monitoring of longitudinal changes in goblet cell populations prior to a pre-term delivery. The strength of the study is the large animal model while weaknesses include sample size relative to the large variability in measures at some time points, histological analytic techniques and the quality of some of the protein stains, and the lack of analysis of post-birth infectious or inflammatory challenges that would enhance relevance to NEC. I largely agree with the conclusions that have been made, and a sensible discussion is provided, however I have some doubt on the robustness of some of the data.

Some specific areas to consider to strengthen the study include:

Comment 1: The number of sheep per group is not clearly stated in Materials & Methods or the Figure legends. I presume each data point is from one sheep with data pooled from the 2 sections analysed – this needs clarification.

Authors response: We agree with the comment of the reviewer. We have added this information in the Materials & Methods section (line 413). Additionally, we have clarified this in the figure legends of figures 1, 2, 3, 4, 5 and 7.

Comment 2: For the ileal cross-sections how were the 2 section areas selected – were only good cross-sections through crypt-villous axes used? Is 2 sections sufficient – ie. how close was the concordance between these sections?  The quality of the histological images and sections selected to demonstrate do not fill me with confidence in the robustness, perhaps contributing to the wide variation in findings at some time points.

Authors response: We thank the reviewer for this valuable comment. Indeed, we have only used good cross-sections through the full crypt-villus axis; this information is added to the Materials & Methods section (line 411-412).
The two sections used contain each approximately 40-50 crypt-villus axes, which means that for each marker we analyzed about 80-100 crypt-villus axes. This is substantially more than analyzing 5 high-power fields at a 200x magnification. Although we agree including more sections would increase test precision, we believe this is a fair amount to reliably measure the markers of interest. To confirm the reliability of our 2 section measurement we used the Spearman–Brown prophecy formula (DOI: 0.1016/j.jclinepi.2017.01.013) that takes into account the interindividual variance (variance between sections in one lamb), the intraindividual variance (variance between all animals) and the number of sections used. The reliability for two sections as estimated with this formula lies between 0.77 (CHOP) and 0.94 (MPO), which is regarded reasonable to good.

Comment 3: The data on goblet cell numbers based on Alcian blue staining (eg. Fig 3 data) will be influenced potentially by goblet cell secretion as larger theca (low secretion) are more likely to appear in sections. This is partially acknowledged in the Discussion of findings but needs to be explicitly acknowledged.  It would have been useful to have goblet cells separately counted in the crypts and villi.

Authors response: The reviewer is absolutely right. We have added “Secretion of mucins from the goblet cells will result in decreased volume of the goblet cell theca and therefore can reduce goblet cell counts on analysis of the AB-PAS staining.” to the discussion (lines 229-230) to more explicitly address this point. Although we understand the question of the reviewer about the goblet cell count in the crypts, we unfortunately cannot determine goblet cell numbers in the crypts, as both Paneth cells and developing goblet cells can positively stain for AB/PAS (https://doi.org/10.1152/ajpgi.00264.2017) and the two cell types cannot be distinguished based on morphology in the developing ovine foetus.

Comment 4: The SPDEF staining is not convincingly specific, all goblet cells should be SPDEF positive so one would predict it should really replicate the Alcian blue data – however, if relying on seeing the theca and SPEDF staining the sectioning angle through individual cells will be impactful. Was there an increase of goblet cell precursors in the base of the crypts?

Authors response: The reviewer is right one would expect that SPDEF positive cells would parallel the Alcian blue data. At present we can only speculate about the observed difference. First, the discrepancy between AB/PAS and SPDEF positive cell counts can be explained by mucus secretion and the subsequent reduction of goblet cell theca size. Second, many SPDEF positive cells were counted in the lower villus region (added to the results section line 165-166) and may thus be developing goblet cells that express SPDEF, but do not have (large) theca yet, which especially in the fetal context might be an important argument. Nevertheless, future studies are needed to determine the reason for the relative high amount of SPDEF-positive cells in this lower villus region in utero. We have elaborated on this apparent discrepancy in the Discussion section, lines 232-239.

Because SPDEF is also a marker for (developing) Paneth cells (https://doi.org/10.1053/j.gastro.2009.06.044), and developing goblet cells and Paneth cells cannot be distinguished in sheep based on cell morphology or cell specific markers, we were unfortunately not able to measure the amount of goblet cell precursors in the base of the crypts.

Comment 5: While I agree there is a high density of CHOP positive cells in the crypt, there are large numbers of villous cells (too numerous for gc) with clear nuclear staining in Fig 8B and D (CHOP is a transcription factor) and this should be described.

Authors response; The reviewer is absolutely right. We added the following sentence to the Results section to address this notion: “The number of CHOP-positive cells was increased in the whole crypt-villus axis following IA LPS, but predominantly highest in the intestinal crypt” Lines (190-191).

Comment 6: The CC3 apoptosis staining is described as being in the transit amplifying cells but looks to be too high in the crypt-villous axis to be in these cells. An alternative possibility is that these are newly differentiating goblet cells.  This is an important distinction that requires further exploration.

Authors response: The reviewer raises an important point. Indeed, we cannot distinguish between transit-amplifying cells and newly formed goblet cells that are terminally differentiating. Unfortunately, to our knowledge, there are at the moment no sheep specific markers available that can distinguish between the two. To address this important notion, we have renamed the TA region as the lower villus region throughout the manuscript and in supplementary figure 3.

Comment 7: Line 268 - BiP is a ubiquitous ER chaperone so is present in health but accumulates in ER stress and it is its disassociation from IRE1 that initiates the UPR. Focus should be on gc.

Authors response: Thank you for this comment. We have rewritten this sentence and changed it into: “BiP is an ER chaperone protein that under physiological conditions prevents inositol-requiring protein 1α (IRE1α) signaling [24]. However, under ER stress, a condition which secretory cells including goblet cells are especially vulnerable to, BiP accumulates and dissociates from IRE1α, thereby initiating the unfolded protein response.” Lines (178-182)

Comment 8: Lines 309-313 – not clear here what is description of current findings and what is being cited from literature – its inappropriately muddled

Authors response: We fully acknowledge that the phrasing of this sentence is somewhat confusing. The findings of early systemic and intestinal inflammation are cited from literature, but refer to studies published from the same experiment. We clarified this in the manuscript in lines 224-228.

Comment 9: Line 325-27 – SPDEF has critical role in mucus production in health and inflammation driving over 100 genes involved in mucin biosynthesis and secretion

Authors response: We have rephrased the sentence to: which suggests that SPDEF has an important role in upregulating mucus production and secretion in mucosal tissue upon inflammation, lines 250-252.

Comment 10: Fig 5 shows apoptosis – claimed to be in TA cells but looks too high in the villi

Authors response: The reviewer raises an important point. Unfortunately, to our knowledge, there are at the moment no sheep specific markers available that can identify the transit amplifying cells. To address this important notion, we have renamed the TA region as the lower villus region throughout the manuscript, including in the discussion of figure 5.

Comment 11: line 194 “of” should be “after” as single LPS exposure with unknown half life?

Authors response: We understand the comment of the reviewer. Previously, we have shown that the half-life time of LPS in the amniotic fluid is relatively long (1.7 days) with LPS concentrations remaining detectable till 15 days after injection (DOI: 10.1067/s0002-9378(03)00758-0). Moreover, we previously showed that systemic and pulmonary immune activation was comparable following 1 or 10 mg of intra-amniotic LPS (DOI: 10.1164/ajrccm.164.6.2103061). Considering the known half-life time, LPS levels remain above this threshold of 1 mg for at least 5 days. Nevertheless, the reviewer is absolutely right that especially for the later time points, LPS exposure will not have lasted the full experimental period. To be as accurate as possible we therefore changed ‘of exposure’ to ‘after exposure’ throughout the manuscript and we have added the information about the LPS half-life time to the Materials & Methods section (lines 347-349).

Comment 12: Line 120 – spelling “Cambridge”

Authors response: Thank you for noticing, we have corrected the spelling in line 366.

Comment 13: Line 156 – spelling “biotin”

Authors response: Thank you for noticing, we have corrected the spelling in line 401.

Reviewer 2 Report

The manuscript submitted for evaluation is at the highest
scientific and editorial level. I find no shortcomings in any of the
elements of the work. Both the design of the study and the extensive
methodology chosen, as well as the meticulous conduct of the experiment
allowed us to obtain reliable results. The statistical methods are not
questionable. The presentation of the results attracts attention with the
rich photographic material. In my opinion, this article is suitable for
publication without corrections.

Author Response

Dear reviewers,

Thank you very much for offering us the opportunity to submit a revised version of our manuscript IJMS-1090160 entitled: "Intestinal goblet cell loss during chorioamnionitis in fetal lambs: mechanistic insights and postnatal implications”

We thank the reviewers for their interest and appreciation of our work. We are grateful for the valuable comments that we feel have improved our manuscript.

Please find enclosed a revised version of the manuscript and a point to point response to the comments of the reviewers.

Reviewer 2:
The manuscript submitted for evaluation is at the highest scientific and editorial level. I find no shortcomings in any of the elements of the work. Both the design of the study and the extensive
methodology chosen, as well as the meticulous conduct of the experiment allowed us to obtain reliable results. The statistical methods are not questionable. The presentation of the results attracts attention with the rich photographic material. In my opinion, this article is suitable for publication without corrections.

Authors response: The authors thank the reviewer for his/her kind words

Reviewer 3 Report

Dear Authors,

This is a very interesting script. The following can improve the logical flow of the methods to the readers:

1- Why apoptosis is chosen as the surrogate marker for Chorioamionitis?
2- Why lipopolysaccharides are chosen as the stress factors for this model? How this correlate with human CA?

Best regards,

Author Response

Dear reviewers,

Thank you very much for offering us the opportunity to submit a revised version of our manuscript IJMS-1090160 entitled: "Intestinal goblet cell loss during chorioamnionitis in fetal lambs: mechanistic insights and postnatal implications”

We thank the reviewers for their interest and appreciation of our work. We are grateful for the valuable comments that we feel have improved our manuscript.

Please find enclosed a revised version of the manuscript and a point to point response to the comments of the reviewers.

Reviewer 3:
Dear Authors,

This is a very interesting script. The following can improve the logical flow of the methods to the readers:

Comment 1: Why apoptosis is chosen as the surrogate marker for Chorioamionitis?

Authors response:
As chorioamnionitis induces fetal gut inflammation and intestinal inflammation is associated with apoptosis of intestinal epithelial cells, we hypothesized that the observed  goblet cell loss after chorioamnionitis may result from apoptosis of intestinal epithelial cells. We further clarified the rationale behind the analysis of apoptosis in the Materials & Methods section at line 142-144.

Comment 2: Why lipopolysaccharides are chosen as the stress factors for this model? How this correlate with human CA?

Authors response:
The reviewer raises an important point. Although LPS is not a living microorganism and specific microorganism-related responses might be missed, this Escherichia coli-derived endotoxin is a potent inducer of inflammation and therefore used to mimic clinical situations of acute inflammation. LPS responses are well-defined and accordingly less heterogenicity in responses is detected (doi: 10.1146/annurev.biochem.71.110601.135414; doi: 10.1016/j.jri.2018.06.029).
Intra-amniotic LPS administration has previously been shown to induce inflammation of the chorion-amnion and provokes influx of inflammatory cells in the amniotic fluid that correlates well with human chorioamnionitis (DOI: 10.1067/s0002-9378(03)00758-0). We have added this rationale to the experimental design paragraph in the Materials & Methods section of the Manuscript (lines 345-347).

Round 2

Reviewer 1 Report

The authors have sensibly addressed the issues I raised, and where appropriate have made changes to the manuscript.